# Effect of the Timing of Hyperbaric Oxygen Therapy on the Prognosis of Patients with Idiopathic Sudden Sensorineural Hearing Loss

**DOI:** 10.3390/biomedicines11102670

**Published:** 2023-09-29

**Authors:** Hsu-Hui Wang, Ya-Ting Chen, San-Fang Chou, Li-Chung Lee, Jia-Hong Wang, Yi-Horng Lai, Hou-Tai Chang

**Affiliations:** 1Hyperbaric Oxygen Therapy Center, Far Eastern Memorial Hospital, Taipei 220216, Taiwan; wangerge@gmail.com (H.-H.W.);; 2Department of Chest Medicine, Far Eastern Memorial Hospital, Taipei 220216, Taiwan; 3Department of Chemical Engineering & Material Science, Yuan Ze University, Taoyuan 320315, Taiwan; 4Department of Medical Research, Far Eastern Memorial Hospital, Taipei 220216, Taiwan; 5Department of Critical Care Medicine, Far Eastern Memorial Hospital, Taipei 220216, Taiwan; 6Department of Healthcare Administration, Asia Eastern University of Science and Technology, Taipei 220303, Taiwan; 7Department of Industrial Engineering and Management, Yuan Ze University, Taoyuan 320315, Taiwan

**Keywords:** hyperbaric oxygen therapy, idiopathic sudden sensorineural hearing loss, pure-tone audiogram

## Abstract

This study aimed to evaluate the effects of hyperbaric oxygen therapy (HBOT) on the hearing recovery of patients with idiopathic sudden sensorineural hearing loss (ISSNHL). The clinical data of 79 patients diagnosed with ISSNHL and treated with HBOT between January 2017 and December 2019 were retrospectively reviewed. The pure tone audiometry (PTA) scores before and after HBOT were recorded. The associations of HBOT efficacy with demographic and clinical characteristics and the duration from disease onset to HBOT administration were determined. The average PTA score was 80.06 ± 25.94 dB before and 60.75 ± 21.26 dB after HBOT; the difference was significant. HBOT improved the hearing of 55.7% of the patients with ISSNHL (defined as an average PTA ≥ 11dB or a final average PTA score below 29 dB). There was a significant inverse relationship between the duration from symptom onset to HBOT administration and PTA score reduction after HBOT, which was adjusted for factors including age, sex, laterality of hearing loss, initial PTA score, reception of intratympanic steroid injections, tinnitus, dizziness, vertigo, diabetes, hypertension, and coronary artery disease. Commencing HBOT at an earlier stage is closely linked to greater improvements in hearing for patients with ISSNHL.

## 1. Introduction

Idiopathic sudden sensorineural hearing loss (ISSNHL) is characterized by a sudden onset of hearing impairment, typically exceeding 30 decibels (dB) over three consecutive frequencies within 72 h. It is not fatal, but it significantly impairs the quality of life of affected individuals. The annual incidence of ISSNHL ranges from 5 to 27 per 100,000 individuals, with approximately 66,000 new cases reported in the United States alone [1]. Previous studies have suggested spontaneous recovery rates of 32–65% [2,3,4].

The pathogenesis of ISSNHL has been debated, and several mechanisms have been suggested [5]. In addition to viral infections and cochlear membrane ruptures, vascular accidents and microvascular injuries have been proposed [6]. Patients diagnosed with ISSNHL have reduced oxygen tension in the vestibular perilymph [7]. Since 1980, the initial treatment approach for ISSNHL has primarily involved the administration of systemic corticosteroids (orally, intravenously, or intramuscularly), as recommended by the national and international consensus guidelines [1,3,5,8]. Other common treatment modalities include systemic vasodilation, intratympanic steroid injections (ITSIs), and hyperbaric oxygen therapy (HBOT).

Hyperbaric oxygen therapy (HBOT) is a widely recognized and accepted medical intervention. It involves the controlled inhalation of pure oxygen within a pressurized environment, typically exceeding 1.4 absolute atmospheric pressure. This treatment significantly elevates the concentration of oxygen in the bloodstream, enhancing its efficient transportation to tissues. It achieves this by surpassing levels achievable through normal respiration, thanks to increased dissolved oxygen concentration in plasma, a phenomenon governed by Henry’s law, which stipulates that the amount of diffused gas in a liquid is directly proportional to its partial pressure above the liquid.

During HBOT, transcutaneous oxygen pressure measurements often demonstrate improved tissue oxygenation [9,10,11]. The heightened oxygen levels in circulation yield a range of physiological effects, including enhanced tissue oxygenation, removal of toxic gases, expedited wound healing and tissue repair, augmented angiogenesis, reduced inflammation through the suppression of inflammatory cytokines like tumor necrosis factor-alpha, interleukin 6, and interleukin 10, increased bactericidal activity of neutrophils, and stimulation of stem cell recruitment [12,13,14,15,16,17,18,19]. Furthermore, increased oxygen pressure yields other advantageous effects on ischemic soft tissues. It promotes energy metabolism preservation, mitigates swelling, bolsters microcirculation and tissue blood supply, encourages fibroblast proliferation and differentiation, elevates collagen synthesis, fosters neovascularization, and enhances the elimination of various bacteria through leukocyte activity [19] These mechanisms have led to extensive discussions on the utility of HBOT in aiding wound healing, particularly in cases of chronic non-healing wounds and severe soft tissue injuries [13,20,21,22,23,24]. Additionally, its benefits in wound healing have been documented in animal models [25]. Beyond wound healing, HBOT finds numerous applications in medicine. These encompass late radiation injury [26], treatment for carbon monoxide poisoning [27], management of retinal artery occlusion [28], amelioration of neuropsychological deficits [29], and assistance in the resolution of intestinal obstruction [30]. Notably, HBOT has also demonstrated efficacy in improving myocardial function in patients suffering from post-COVID-19 syndrome [31]. Extensive clinical evidence has demonstrated the safety of HBOT; the most common adverse effect of HBOT is barotrauma of the middle ear. Severe complications, such as oxygen toxicity of the central nervous system, are rare and typically reversible [32].

ISSNHL has been investigated as a potential indication for HBOT since the 1970s, guided by the hypothesis that HBOT can alleviate cochlear hypoxia [10,11]. Besides, the mechanism of HBOT in wound healing can also be applied to improving ISSNHL. In a systematic review conducted by Joshua et al. in 2021 [33], HBOT as part of combination treatment was associated with improved hearing recovery in patients diagnosed with ISSNHL. A meta-analysis conducted by Rhee et al. in 2018 further supported the inclusion of HBOT as a viable treatment option for ISSNHL, especially in patients with severe-to-profound hearing loss at the initial assessment. Another meta-analysis demonstrated that HBOT could be a reasonable addition to standard medical therapy for ISSNHL when administered as salvage treatment for an extended duration [34]. In a 2012 Cochrane review, the use of HBOT resulted in a significant improvement in the hearing outcomes of patients with ISSNHL [35]. Since 2011, the Undersea Hyperbaric Medicine Society (UHMS) has included ISSNHL in its list of approved indications for HBOT. Additionally, the European Committee on Hyperbaric Medicine (ECHM) issued a type 1 recommendation endorsing the use of HBOT as a treatment option for ISSNHL in 2016 [36]. The 2019 practice guidelines of the American Academy of Otolaryngology recommended HBOT as an optional initial treatment within two weeks of symptom onset in patients with ISSNHL. HBOT has also been suggested as salvage therapy for patients who do not respond to initial pharmacological therapy within one month [1]. Nevertheless, the optimal timing for initiating HBOT in patients with ISSNHL is yet to be established, given that there are limited studies on it and their relevance is limited [37,38,39].

In our daily practice, we have observed that the chances of hearing loss recovery in patients with ISSNHL increase when they receive HBOT at an earlier stage from the onset of symptoms. However, the question arises: what constitutes ‘early’ in this context, and does the timing of HBOT indeed influence the recovery rate? Additionally, we have noted instances where patients with ISSNHL were referred for HBOT evaluation as a salvage therapy only after other treatment modalities had failed to improve hearing loss for a duration of four weeks or longer from symptom onset. Unfortunately, our clinical observations suggest limited benefits in such cases.

To bridge this knowledge gap, we initiated a retrospective study to investigate the potential impact of the timing of HBOT on the prognosis of patients with ISSNHL. Our objective was to validate our hypothesis: that earlier administration of HBOT would lead to more significant improvements in hearing recovery.

## 2. Materials and Methods

### 2.1. Patients

This retrospective study analyzed the data of patients diagnosed with SSNHL who received HBOT at the Far Eastern Memorial Hospital (FEMH) between 2017 and 2019. The inclusion criteria for this study involved selecting patients aged 20 years or older, diagnosed with ISSNHL, and who received HBOT at FEMH. The exclusion criteria for this study included age of <20 years, known hearing loss etiology, bilateral or previous ipsilateral hearing loss, HBOT initiation beyond 180 days after symptom onset, insufficient follow-up data for pure-tone audiometry (PTA), fewer than three HBOT sessions, and lack of audiogram data beyond one week after the initial HBOT session. A cohort of 79 patients who satisfied the inclusion criteria was included in the analysis. Along with HBOT, most patients (all except one) received steroid treatments, such as systemic steroids (oral or intravenous) or ITSIs, in otorhinolaryngology clinics. Ethical approval was obtained from the Far Eastern Memorial Hospital Ethics Committee (no. FEMH-IRB-110231-E, approved on 26 October 2021). Informed consent is waived under FEMH Ethnics Committee agreement due to the observational nature of the investigation. No new interventions or interactions with patients are involved, as the data has already been collected for clinical purposes. Additionally, the retrospective design ensures that patient identities remain anonymized, and confidentiality is maintained throughout the research process.

### 2.2. Hyperbaric Oxygen Therapy (HBOT)

The therapy center at the FEMH used the HAUX Starmed 2200 hyperbaric chamber, manufactured by Haux-Life-Support GmbH in Karlsbad, Germany, for HBOT administration. Each HBOT session was conducted at a pressure of 2.5 absolute atmospheres and lasted for 120 min. This protocol comprises a 15-min descent time, a 90-min bottom time, and a 15-min ascent time. Patients received pure oxygen during the bottom and ascent phases, with two 5-min air breaks during the bottom phase to mitigate the risk of oxygen toxicity, and the last 3 min of the ascent phase also involved a switch to air. Each session’s duration of pure oxygen breathing is estimated to be 92 min. The treatment sessions were conducted daily. Ten to 20 HBOT sessions have been recommended by previous studies, but the actual number of sessions varied for each patient based on their needs and response to treatment.

### 2.3. Audiogram Evaluation

PTA was used to evaluate hearing improvement in individuals presenting with ISSNHL symptoms. PTA scores in decibels (dB) were determined for the frequencies of 250, 500, 1000, 2000 and 4000 Hz. The average PTA score was calculated and is reported in this study. The initial PTA score was determined based on the most recent audiogram obtained before HBOT initiation. The outcomes were recorded following the completion of the HBOT sessions. In this study, the overall improvement was defined as an average PTA improvement exceeding 10 dB or a final average PTA score below 29 dB.

### 2.4. Data Analysis

The main aim of this retrospective study was to examine the correlations between the efficacy of HBOT and various factors. These factors included the duration from the onset of symptoms to the initiation of HBOT, demographic characteristics, clinical features (such as the side of hearing loss, initial PTA, presence of comorbidity, and associated symptoms), and use of systemic corticosteroids or ITSIs.

The Kolmogorov-Smirnov test was used to assess the normality of the numerical data. Normally distributed data are expressed as mean ± standard deviation, whereas non-normally-distributed data are represented as medians (Q1, Q3). Categorical data were presented as frequencies and percentages. The paired-sample *t*-test or Wilcoxon signed-rank test was used to compare the pre-and post-HBOT continuous data.

To investigate the differences in hearing improvement across various durations from symptom onset to HBOT, a combination of one-way ANOVA/Welch and post-hoc Games-Howell analyses were conducted. Univariate and multivariate linear regression were used to assess the impact of the duration from symptom onset to HBOT on hearing improvement while considering other covariates.

The threshold for determining statistical significance was set at *p* < 0.05. All statistical analyses were performed using SPSS software version 22.0 (SPSS Inc., Chicago, IL, USA).

## 3. Results

The study enrolled 79 participants. Table 1 summarizes their demographic and clinical characteristics. Seventeen (21.5%), 25 (31.6%), 20 (25.3%), and 17 (21.5%) received HBOT ≤ 7, 8–14, 15–30, and >30 days after symptom onset, respectively. Seventy-eight (98.7%) received systemic steroids within one week, and 49 (62.0%) received ITSIs.

Table 2 compares the hearing loss grades of the patients diagnosed with ISSNHL before and after receiving HBOT. The average PTA score for hearing loss was 60.75 ± 21.26 dB after and 80.06 ± 25.94 dB before HBOT; the difference was significant (*p* < 0.001).

Table 3 presents the data on the effectiveness of HBOT in improving the PTA score. The median PTA improvement was 13 dB (range, −10–60 dB). Among the patients, 55.7% experienced overall improvement, defined as an average increase in the PTA score exceeding 10 dB or a final average PTA score below 29 dB. Additionally, 16.5% of the participants achieved complete recovery with a PTA score of 29 dB or lower.

Linear regression showed an inverse correlation between the duration from symptom onset to the initiation of HBOT and the reduction in PTA score (ΔPTA) after HBOT (Beta = −0.293, *p* < 0.001) (Table 4). Compared with the patients who received HBOT within 7 days after symptom onset (Period 1), those who received it 8–14 (Period 2), 15–30 (Period 3), and >30 (Period 4) days had lower mean ΔPTA values. This suggests that a delay in the initiation of HBOT after symptom onset may impede hearing recovery.

Incorporating a multivariate linear regression analysis, we accounted for several potential variables that could potentially influence the observed improvement. These encompassed factors such as age, sex, laterality of hearing loss, initial PTA score, receipt of ITSI, as well as coexisting conditions including tinnitus, dizziness, vertigo, diabetes, hypertension, and coronary artery disease. The analysis revealed a noteworthy correlation between the efficacy of HBOT and the time elapsed from the onset of symptoms to the commencement of treatment (Beta = −0.296, *p* < 0.001), even subsequent to meticulous adjustment for these aforementioned variables (delineated in Table 5).

In these 79 patients with idiopathic sudden sensorineural hearing loss and receiving HBOT, the most common complication is eardrum barotrauma, whose severity is often mild and self-limited. They did not experience CNS toxicity during HBOT.

## 4. Discussion

The primary objective of this retrospective study was to establish the ideal timing for initiating HBOT in patients diagnosed with ISSNHL. To achieve this, we examined the influence of the timing of HBOT initiation on hearing improvement. Our findings revealed a significant negative correlation between the duration from symptom onset to the commencement of HBOT and the magnitude of improvement in the PTA score. These results strongly indicate that initiating HBOT earlier is associated with greater hearing improvement in patients with ISSNHL. Notably, this observation is a novel contribution to the existing literature.

The pathophysiology of ISSNHL is closely related to the condition of the inner ear [40]. In a systemic review conducted by Yamada et al. [6], ISSNHL’s development may result from disruptions in cochlear circulation and cochlear damage caused by infections or inflammation stemming from other etiologies. Maintaining normal cochlear microcirculation is crucial for the inner ear since sensory hair cells are susceptible to hypoxia, which can impact hearing thresholds. [41] In an animal study led by Kerstin Lamm et al. [42], it was observed that noise-induced cochlear hypoxia could potentially be compensated for by HBOT. This suggests that HBOT might improve inner ear hypoxia, thereby improving hearing loss. Furthermore, inflammation-induced cochlear damage plays a significant role in hearing loss [43]. The anti-inflammatory effects of HBOT [19] may also contribute to its ability to reduce cochlear inflammation by decreasing inflammatory mediators, thereby improving hearing loss.

Goto et al. [44] observed that initiating HBOT within two weeks of symptom onset led to better hearing recovery, especially when combined with steroid treatment. Similarly, a retrospective study by Yıldırım et al. [37] demonstrated that initiating HBOT within the first 14 days of diagnosis positively influenced the prognosis of patients with ISSNHL. A separate retrospective study conducted by Muzzi et al. [45] demonstrated the potential of HBOT in improving pure-tone hearing thresholds in patients with ISSNHL who did not respond to medical therapy alone. This study further demonstrated that favorable outcomes were more likely when HBOT was administered within 30 days after symptom onset. Kayalıet al [46] reported follow-up outcomes indicating that salvage HBOT three weeks after the onset of ISSNHL symptoms was not significantly effective in patients who did not respond to corticosteroid treatment. Chin et al. [47] reported a cutoff of 12.5 days as the optimal timing for initiating HBOT for patients with ISSNHL. This conclusion was based on receiver operating characteristic (ROC) analysis. Therefore, early initiation of HBOT may lead to better treatment outcomes for ISSNHL, consistent with our analysis and findings from the existing literature. Furthermore, the results of our study indicate that the optimal outcome of HBOT for ISSNHL is not confined to a specific time window but rather occurs when treatment is initiated as early as possible from the onset of symptoms. Likewise, Mariani et al.’s study raised questions regarding the use of HBOT and ITSI as salvage therapies [48].

Our study also revealed a significant positive correlation between the severity of initial hearing loss and improvement in PTA scores. Individuals with more severe hearing loss at baseline may experience marked benefits from HBOT. In another retrospective study by Ajduk et al., significant improvements in hearing thresholds were observed across all frequencies for patients with hearing loss exceeding 61 dB who received salvage HBOT after failed steroid treatment for ISSNHL [49].

Most patients in our study received systemic corticosteroids within one week of experiencing ISSNHL symptoms. Therefore, recovery from hearing impairment in our study was not affected by the duration from symptom onset to the administration of systemic steroids.

Based on our observations, ITSIs had no significant impact on hearing improvement in patients diagnosed with ISSNHL receiving HBOT. A meta-analysis conducted by Sialakis et al. [50] also indicated that there was no significant difference between the efficacies of intratympanic and systemic steroid therapies. Further randomized prospective studies are required to improve our understanding of these results. Nonetheless, ITSIs remain a viable treatment option for patients with contraindications to systemic steroids when considering HBOT.

Comorbidities such as diabetes mellitus, hypertension, and coronary artery disease did not have a significant effect on the overall improvement in hearing loss in patients with ISSNHL. This implies that individuals with these comorbidities may benefit from HBOT as a viable treatment option for ISSNHL.

We obtained audiograms within 180 days from the onset of ISSNHL to monitor changes in the PTA scores. Based on our patient data, we observed that hearing changes tended to stabilize approximately one month after ISSNHL onset. Moon et al. [51] suggested that the prognosis of ISSNHL can be predicted approximately two weeks after the initiation of treatment. They also recommended follow-up with audiograms for at least two months after treatment for patients with incomplete or delayed hearing improvement. Liu et al. [52] reported that the curative effect of treatment for ISSNHL stabilized after two months. Therefore, we consider a two-month follow-up period for audiograms appropriate for evaluating treatment efficacy.

Several other studies have explored the optimal timing of HBOT for ISSNHL [37, 46, 48]. We believe our study adds a valuable contribution to the existing literature for several compelling reasons:

1. **Novel Inverse Relationship**: Our study distinguishes itself by being among the few to elucidate an inverse relationship between the duration of receiving HBOT from the onset of symptoms and the improvement of hearing loss. Unlike most papers on this topic that simply identify specific time windows, we delve into the dynamic relationship between timing and outcomes.

2. **Comparable Patient Numbers**: Our study ensures comparability by maintaining similar patient numbers across different durations. This approach strengthens the validity of our comparisons regarding the effects of treatment duration.

3. **Clinical Realism**: We acknowledge the clinical reality where some patients turn to HBOT as a salvage therapy after other treatments have failed. Drawing from our observations, we advocate for early HBOT intervention, as we have noticed that delays in treatment initiation may result in less favorable outcomes.

These distinctive features of our study enrich the current body of literature on HBOT for ISSNHL, providing valuable insights for both researchers and clinicians.

## 5. Strength & Limitation

This study had several limitations primarily attributed to its retrospective design and relatively small sample size. Additionally, the treatment regimens, including systemic corticosteroids and ITSIs, administered to each patient were not standardized but based on individualized assessments. This may have influenced the outcomes of the study. To validate these findings, further studies involving larger samples and more rigorous study designs are required.

## 6. Conclusions

This study underscores the critical role of the timing of HBOT initiation in determining the prognosis of patients with ISSNHL. Our findings strongly support the notion that commencing HBOT at an earlier stage is closely linked to greater improvements in hearing for patients with ISSNHL. This discovery holds substantial implications for clinical decision-making, emphasizing the importance of early HBOT administration as an integral part of the treatment for ISSNHL, rather than solely as a last-resort salvage therapy following unsuccessful attempts with other treatment modalities. Future prospective studies with larger sample sizes are warranted to validate these results and gain deeper insights into the therapeutic advantages of HBOT in ISSNHL.

## Figures and Tables

**Table 1 biomedicines-11-02670-t001:** The demographic and clinical characteristics of the ISSNHL patients with HBOT. (N = 79).

Characteristic	Statistics	Range, Min–Max
Age, mean ± SD	52.61 ± 12.82	25–86
Sex, n (%)		
Male	42 (53.2%)	
Female	37 (46.8%)	
Side of Hearing Loss, n (%)		
Right	37 (46.8%)	
Left	42 (53.2%)	
Time interval between onset of symptoms and HBOT, day, median (Q1, Q3)	13 (8, 25)	1–132
≤7 days	17 (21.5%)	
8–14 days	25 (31.6%)	
15–30 days	20 (25.3%)	
≥31 days	17 (21.5%)	
Systemic steroid	78 (98.7%)	
ITSI Reception	49 (62.0%)	
Associated symptoms		
Tinnitus	74 (93.7%)	
Dizziness	30 (38.0%)	
Vertigo	20 (25.3%)	
Comorbidity		
Diabetes	15 (19.0%)	
Hypertension	22 (27.8%)	
Coronary artery disease	5 (6.3%)	

The normality of numerical data was tested by Kolmogorov-Smirnov test. Normally distributed. data are expressed as mean ± standard deviation while non-normally distributed data was expressed as median (Q1, Q3). Categorical data was expressed as frequency and percentage.

**Table 2 biomedicines-11-02670-t002:** Comparison of the PTA score/Grade of ISSNHL patients before and after HBOT.

Variables	Before HBOT	After HBOT	*p*-Value
Averaged PTA score dB	80.06 ± 21.26	60.75 ± 25.94	<0.001 a,*
PTA Grade			
≤29 dB	0 (0%)	13 (16.5%)	
30–60 dB	12 (15.2%)	25 (31.6%)	
61–80 dB	30 (38.0%)	26 (32.9%)	
≥81 dB	37 (46.8%)	15 (19.0%)	

a Paired-Samples T Test compared the difference of PTA score before and after HBOT. * Indicates the results reached a significant difference between before and after HBOT (*p* < 0.05).

**Table 3 biomedicines-11-02670-t003:** The efficacy of HBOT on improvement of PTA score.

Variables	Statistics	Range, Min–Max
ΔPTA score before and after HBOT (dB), median (Q1, Q3)	13 (2, 36)	−10–60
Overall Improvement, n (%)		
Yes	44 (55.7%)	
ΔPTA = 11–30 dB	18 (22.8%)	
ΔPTA ≥ 31 dB	13 (16.5%)	
Final PTA score ≤ 29 dB	13 (16.5%)	
No		
ΔPTA ≤ 10 dB	35 (44.3%)	

**Table 4 biomedicines-11-02670-t004:** Evaluation of the effect of time intervals between the onset and HBOT on PTA score improvement by univariable linear regression analysis. N = 79.

Variables	Beta	*p*-Value
Independent		
ΔPTA score before and after HBOT (dB)		
Dependent		
Intervals between the onset and HBOT, days	−0.293	<0.001 *
Intervals between the onset and HBOT, periods	−9.658	<0.001 ^a,^*
Period 1: ≤ 7 days	Ref	-
Period 2: 8–14 days	−14.6 ^b^	0.009 ^b,^*
Period 3: 15–30 days	−19.03 ^b^	0.002 ^b,^*
Period 4: ≥ 31 days	−31.35 ^b^	<0.001 ^b,^*

Represents the difference in PTA score between before and after HBOT. * Indicates the results reached a significant difference (*p* < 0.05). ^a^ shows the statistical result of the association of the difference of PTA score before and after HBOT (ΔPTA) with the periods between the onset and HBOT. ^b^ shows the statistical results comparing the indicated group with the Ref group period 1.

**Table 5 biomedicines-11-02670-t005:** Analysis of the efficacy of time intervals between the onset and HBOT on PTA score improvement adjusted for other potential variables by multivariate analysis.

Variables	Beta	*p*-Value
Outcome		
ΔPTA score before and after HBOT (dB)		
Variables		
Time intervals between the onset and HBOT, days	−0.296	<0.001 *
Age	0.056	0.795
Gender, Male to female	−2.315	0.598
Side of Hearing Loss, Right to Left	−4.688	0.291
PTA score before HBOT	0.168	0.150
ITSI Reception	2.584	0.589
Tinnitus	5.536	0.539
Dizziness	−11.338	0.063
Vertigo	7.732	0.289
Diabetes	10.312	0.100
Hypertension	−9.200	0.136
Coronary artery disease	−10.947	0.302

Δ represents the difference PTA score before and after HBOT. * Indicates the results reached a significant difference (*p*< 0.05).

## Data Availability

All data generated or analyzed in this study are included in the published article. Data are available upon reasonable request from the corresponding authors.

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
