# Peer review of "Effect of the Timing of Hyperbaric Oxygen Therapy on the Prognosis of Patients with Idiopathic Sudden Sensorineural Hearing Loss"

_biomedicines, 2023, doi:10.3390/biomedicines11102670_

Round 1

Reviewer 1 Report

The manuscript accounts of a study evaluating the effects of hyperbaric oxygen therapy on the hearing recovery of patients with idiopathic sudden sensorineural hearing loss (ISSNHL). The group studied was limited (79 patients) so the study must be regarded as preliminary, nevertheless may be a starting point for a more extensive study. The conditions and duration of single treatment was constant (2.5 atmospheres, 120 min). In spite of non-uniformity of the basic treatment, application of hyperbaric therapy stimulated hearing recovery. The study demonstrated that the positive effect is inversely correlated with the time of onset of hyperbaric oxygen therapy after the symptom onset, which is a valuable confirmation of results of previous studies. The correlation between the severity of  initial hearing loss and improvement in PTA scores is also interesting.

Remarks:

It is not obvious from the description of Methods (2.2) if the therapy was based on applying increase pressure of air or of pure oxygen; it could be stated, for the sake of clarity.

I suggest a short deepened discussion possible mechanisms of the beneficial effects of hyperbaric oxygen therapy.

The authors attribute the positive effects of hyperbaric oxygen therapy to better oxygenation of blood, exceeding those achievable through normal respiration. Indeed, according to the Henry’s law, the amount of oxygen dissolved physically in a fluid is proportional to its pressure above the fluid. However, oxygen is delivered to tissues predominantly in the form bound to hemoglobin and hemoglobin in the blood leaving the lungs is almost totally saturated with oxygen so the increase in hemoglobin saturation by increasing oxygen pressure is minimal.

Perhaps there can be also other potential mechanisms of the beneficial effect of hyperbaric oxygen therapy? One idea I could propose is connected to the beneficial effects of oxygen on wound healing. If the hearing loss is due to some (micro)wounds in the ear, perhaps these wounds heal better under elevated oxygen pressure? I am sure the authors have other interesting ideas.

Are there any experiments allowing to discriminate between the effects of increased pressure and increased oxygen pressure?

Author Response

Please see the attachement , 

Reviewer 2 Report

The main aim of this retrospective study was to determine the optimal timing for hyperbaric oxygen therapy (HBOT) initiation for patients diagnosed with idiopathic sudden sensorineural hearing loss (ISSNHL). The Authors showed that early HBOT initiation is associated with improved hearing recovery.

The paper is well-written, and logically structured, and provides interesting results with potential clinical importance. The Authors have presented sufficient data. The appropriate tables have been provided. The methods are generally adequately described. The authors used appropriate statistical methods. The conclusions are consistent with the presented evidence and arguments.

 Nevertheless, there are some improvements that could be made, as follows:

1.     In the introduction lines 48-54 please add more information for readers about where HBO is used in medicine :

-improving the treatment of leg chronic wounds [doi: 10.17219/acem/92304. PMID: 30238703]; late radiation injuries and carbon monoxide poisoning [doi: 10.3390/biom11081210]; neuropsychological deficits [doi: 10.1007/s11065-021-09500-9.]; HBOT promotes left ventricular systolic function recovery in patients suffering from post COVID-19 condition [doi: 10.1038/s41598-023-36570-x.]

2.     and its potential mechanisms:

the increased oxygen pressure effects ischemic soft tissues by improving the preservation of energy metabolism and reducing swelling; these factors improve the microcirculation and blood supply to tissues;  stimulation of fibroblast proliferation and differentiation, increase collagen synthesis, improve neovascularization, and destruction of different kinds of bacteria by leukocytes [https://doi.org/10.1007/s10973-011-1934-6;  doi: 10.3390/medicina57090864; doi: 10.3390/biom11081210;]

 Did you observe any side effects in your patients during HBO treatment?

Reviewer 3 Report

The manuscript was prepared very well. The introduction section justifies the purpose of the study. I congratulate the authors for the preparation of the manuscript

I would like to congratulate the authors for the structure of the manuscript and all the research carried out. It is highly publishable. However, there are some concerns, in part important, so the review articles need revision, see below.

Introduction

-        Why is this study considered relevant?

-        Why is this study necessary?

-        add some more of the applications of hyperbaric oxygen therapy

-        what is your hypothesis?

Methods and Results

-        It is one of the strong parts of the manuscript, these excellently described

Discussion

·       Include a section on strengths / limitations.

·       What mechanisms of action support these findings?

·       What does this article contribute to, the authors should make their own assessment and include their own discussion of the results shown in the manuscript?

Conclusion

In the Conclusion section, state the most important outcome of your work. Do not simply summarize the points already made in the body — instead, interpret your findings at a higher level of abstraction. Show whether, or to what extent, you have succeeded in addressing the need stated in the Introduction (or objectives).

Requires substantial changes and must be reviewed by someone native

Round 2

Reviewer 3 Report

The authors have made the proposed suggestions, however, they must make the 2 proposed suggestions:

They should be explicit about what this article contributes, for this I recommend that the authors should make their own assessment and

In addition, they should include some mechanism or idea (reference) that justifies their results or comparison with similar studies.

You should check typo errors and some expressions

Author Response

Thank you once again for your valuable input.
